# CatBoost-Based Automatic Classification Study of River Network

**Di Wang** and **Haizhong Qian** *

Institute of Geospatial Information, Information Engineering University, Zhengzhou 450052, China; lianganwuci@gmail.com
* Correspondence: haizhongqian@163.com

**Abstract:** Existing research on automatic river network classification methods has difficulty scientifically quantifying and determining feature threshold settings and evaluating weights when calculating multi-indicator features of the local and overall structures of river reaches. In order to further improve the accuracy of river network classification and evaluate the feature weight, this paper proposes an automatic grading method for river networks based on ensemble learning in CatBoost. First, the graded river network based on expert knowledge is taken as the case; with the support of the existing case results, a total of eight features from the semantic, geometric, and topological aspects of the river network were selected for calculation. Second, the classification model, obtained through learning and training, was used to calculate the classification results of the main stream and tributaries of the river reach to be classified. Furthermore, the main stream river reaches were connected, and the main stream rivers at different levels were hierarchized to achieve river network classification. Finally, the Shapley Additive explanation (SHAP) framework for interpreting machine learning models was introduced to test the influence of feature terms on the classification results from the global and local aspects, so as to improve the interpretability and transparency of the model. Performance evaluation can determine the advantages and disadvantages of the classifier, improve the classification effect and practicability of the classifier, and improve the accuracy and reliability of river network classification. The experiment demonstrates that the proposed method achieves expert-level imitation and has higher accuracy for identifying the main stream and tributaries of river networks. Compared with other classification algorithms, the accuracy was improved by 0.85–5.94%, the precision was improved by 1.82–9.84%, and the F1_Score was improved by 0.8–5.74%. In this paper, CatBoost is used for river network classification for the first time, and SHAP is used to explain the influence of characteristics, which improves the accuracy of river network classification and enhances the interpretability of the classification method. By constructing a reasonable hierarchy, a better grading effect can be achieved, and the intelligence level of automatic grading of river networks can be further improved.

**Keywords:** river network automatic classification; main stream recognition; SHAP framework; ensemble learning

## 1. Introduction

Water systems are a significant element of maps, with river networks being an essential type of water system. The use of intelligent methods for automatic map generalization can reduce uncertainty and effectively improve the efficiency of the cartography of river network elements. From the point of view of cartography, the classification of river networks is the basis of river network cartography generalization, which can simplify the complex river network system into different levels of river reaches and make the structure of river networks clearer and easier to understand and master.

The watercourse connections on a map are composed of a river network, rivers, and river reaches. Tree-shaped (dendritic), trellis, and feather-shaped river networks have

a clear hierarchical structure, with a main stream and several levels of tributaries in a 'parent–child' structure [1], meaning that river classification is a fundamental and vital task for generalizing river networks. Current studies on the automatic classification of river networks mainly focus on tree-shaped river systems; relevant studies include those by Tan et al., who used the weighted average planning method with maximum affiliation in multi-criteria decision-making to consider the spatial relationship and spatial attribute factors of the river network main stream [2]. Guo and Huang automatically inferred the relationship between the main stream and tributaries of a tree-shaped river network based on the flow direction of river reaches and the cumulative number and length of river reaches at node branches [3]. Zhai and Xue structured rivers and planar water systems and selected mainstreams based on the river, including angles, river reach depths, and other characteristics [4]. Li et al. constructed tree-shaped river network stroke connections and structured rivers hierarchically by considering features such as river semantics, length, and angle [5]. Previous studies have extracted various features of river networks, but they mainly focus on using traditional mathematical methods and expert experience knowledge to limit the threshold of different features, which has a significant subjective impact. In the field of river network cartographic generalization, river network classification plays an important role in pattern recognition, selection, simplification, hydrological analysis, river ecological analysis, and river management planning. Although traditional mathematical methods have certain advantages in river network generalization, machine learning, and deep learning methods have obvious advantages in data processing efficiency, model accuracy, modeling process optimization, and data quality requirements [6,7]. In recent years, there has been much research on using machine learning and deep learning methods to deal with river network cartography. It used Support Vector Machines (SVM) [8,9], graph neural [10–12], classification trees [13], deep neural [14], and other methods for pattern recognition of river network structure types. Back Propagation Neural Network (BPNN) and decision tree (DT) [15,16], naïve Bayes [17], hybrid coding [18], genetic algorithm [19], and GraphSAGE [20] were used for river selection. In river network simplification, the DP algorithm [21–25] and Li-Openshaw [26], self-organizing maps (SOM), and Genetic Algorithms (GA) [27–30] were used to achieve line simplification. In hydrological analysis, Convolution Neural Networks (CNN) and Long Short-Term Memory (LSTM) were used for hydrological simulation [31] and water level prediction [32], and deep learning was used for hydrological model calibration [33]. In terms of river ecological governance and analysis, machine learning was used to estimate river ecological status [34], predict watershed carbon emissions [35], and use the Gated Recurrent Unit (GRU) to forecast river discharge [36].

Ensemble learning, as a technical framework, is an important method in machine learning. It can combine and construct the basic model according to different ideas and combine multiple learners to complete the task, so as to achieve better purposes. It has been extensively studied and applied in the fields of classification and prediction [37,38], object detection [39,40], and text classification [41,42]. However, there is currently a lack of research on the automatic classification of river networks using the ensemble learning method. As an important ensemble learning method, the CatBoost algorithm is suitable for processing machine learning datasets of various sizes. It can automatically deal with problems such as categorical features and data imbalances and obtain better performance by adjusting and optimizing hyperparameters. The application of CatBoost in hydrology includes river modeling and cartography [43], flow prediction [44], and flood prediction [45,46], and has achieved good research results. In this paper, CatBoost is used to simulate the decision-making process of expert classification and enrich and achieve intelligent classification of river networks.

In summary, this paper proposes a CatBoost algorithm for automatically classifying river networks. The algorithm was used to calculate eight semantic, geometric, and topological features of river networks. The classification model obtained by training and testing a complex river network classified by experts could identify the main stream and tribu-

taries and achieve automatic classification. Furthermore, to address problems with the ensemble learning algorithm related to complex structures and poor interpretability, the Shapley additive explanation (SHAP) method was introduced to quantify the influence of each feature term on the results and further improve the transparency of CatBoost black-box learning.

## 2. Main Principles of the Algorithm

### 2.1. CatBoost Algorithm

CatBoost is a gradient-boosting decision tree (GBDT) framework with an oblivious tree as the base learner and fewer parameters. It supports categorical variables while achieving high accuracy. Trains a series of learners serially using the boosting method and accumulates the outputs of all learners as a result [47], thereby improving the accuracy and applicability of the algorithm. For a given training set with n samples $D\left\{(X_i, Y_i)_{i=1,2,\dots,n}\right\}$, where $X_i = (x_i^1, x_i^2 \dots, x_i^m)$ denotes the m-dimensional input features and $Y_i \in R$ denotes labeled values. The strong learner generated after training is $F_{k-1}$, and the training goal in the next round is to obtain a tree $t_k$ from the CART decision tree set T to minimize the expectation $E(\cdot)$ of the loss function $L(\cdot)$. The parameter $t_k$ is calculated as follows:

$$t_k = \text{argminE}L(y, F_{k-1}(x) + t(x)) \tag{1}$$

where $(x, y)$ are test samples independent of the training set. The GBDT uses the negative gradient of the loss function to fit the trained CART decision tree $t_k$, and the final model M shown in Equation (2) is obtained from the initial weak learner $M_0$ and the *n*-th round of the training step size $a_n$ after $N$ iterations:

$$M = M_0 + \sum_{k=1}^{N} a_n t_k \tag{2}$$

CatBoost makes some improvements based on the traditional GBDT and has the following innovations compared with other boosting algorithms: 1. CatBoost introduces order boosting against the noise points in the training set [48]; 2. CatBoost automatically converts categorical features to numerical features by drawing on the Ordered Target Statistics (Ordered TS) method, which increases the direct support for categorical features; 3. CatBoost uses categorical features, which significantly enriches the feature dimension; and 4. it uses a fully symmetric tree as the base model and implements the same splitting criterion for each layer [49], which improves the stability and prediction speed.

### 2.2. Main Stream and Tributary Case Feature Extraction

The basic classification criteria based on expert knowledge are based on the national river classification criteria. In a water system, the river that flows directly into the primary main stream is called the secondary mainstream, the river that flows directly into the secondary main stream is called the tertiary mainstream, and so on. For smaller rivers, based on the existing classification results, the connectivity, intersection angle, and length of the river are mainly considered to determine whether the river reach belongs to the main stream or tributary. Since the main stream and tributary are relative, especially when the main tributary is determined with the river reach as the basic unit, it is mainly to judge which river reach at the intersection is more in line with the main stream characteristics. Through an analysis of related literature, it was determined that the upstream and downstream flow directions of main stream rivers usually maintain a straight line, which satisfies the '180° assumption' [50]. First-class main stream rivers not only satisfy the 180° assumption at the confluence point but are also generally the longest rivers in length [51]. The main stream generally has the largest number of tributaries [2], and the names of the rivers that reach main stream rivers are generally consistent [16]. This study combined ex-

isting knowledge and experimental data features to select eight indicators as case feature terms, including semantic, geometric, and topological features. (Table 1).

**Table 1.** River reach classification case features.

| Case Features | | Description |
|---|---|---|
| Semantic feature | Consistency of semantic(C_S) | Reflects the semantic attribute characteristics of adjacent river reaches and assigns a value of 1 to the downstream reach if the name of the downstream reach is the same as that of the upstream reach and 0 to the downstream reach if it is not the same |
| Geometric features | Curvature of reaches(C_R) | According to the morphological characteristics of the river reach, the greater the curvature of the river reach, the smaller the probability of being classified as the mainstream. |
| | Length of reaches(L_R) | The longer the length of the river reach, the greater the probability of being classified as the mainstream. |
| | Angle of intersection(A_I) | Search for river confluence points, and calculate the intersection angle by the first and last endpoints of the river reach; the closer the angle is to 180°, the higher the probability of being classified as the mainstream. |
| Topological features | Number of upstream reaches(N_R) | The higher the number of adjacent reaches upstream of the river reach to be classified; the higher the probability of it being classified as the mainstream. |
| | Depth of reaches(D_R) | Construct a tree structure with leaf nodes with a depth of 1 to calculate the depth of the river reach; the greater the depth, the greater the probability of being classified as the mainstream. |
| | Number of upstream river sources(N_S) | Trace upstream to the source; the greater the number of sources, the greater the probability of being classified as the mainstream. |
| | Maximum upstream length(M_U) | The maximum length of the river traced upstream to its source; the greater the maximum length of the river, the higher the probability of being classified as the mainstream. |

### 2.3. Automatic Classification Strategy for River Networks Based on the CatBoost Model

Based on a three-level structure of river networks, rivers, and river reaches, each river in the river network was divided into river reaches from confluence points. A data model based on river reaches was established to identify the relationship between the main stream and tributaries of the river from the mouth to the upper reaches. Thus, the river is constructed through the main stream and tributary relationships of the river reach, and the automatic hierarchical classification of the river network is completed.

The difficulty and focus of identifying the main stream and tributaries of a river reach lie in scientifically selecting relevant indicators and setting indicator weights appropriately. From the perspective of case-supported learning, cases are acquired from the existing expert pool, and further comprehensive knowledge implied in the cases is obtained. Machine learning algorithms are used to generate classification models through training and testing to guide the identification of the main stream and tributaries of new river networks. The CatBoost classification model is easy to implement and has strong classification performance and high accuracy. As a result, the CatBoost model was selected in this study to classify the mainstreams and tributaries of river reaches. The specific steps are as follows.

Step 1: Obtain the expertly classified river network, split it into several river reaches with classifications and main stream and tributary identification at confluences, and calculate the feature terms of the classified river reaches.

Step 2: Input the extracted case feature terms into the CatBoost classification model for training and testing, and generate the final classification decision model after parameter optimization.

Step 3: Prepare the river network to be classified, extract the features of river reaches, use the above classification model to identify the main stream and tributaries, and hierarchically structure the main stream and tributaries to achieve river network classification.

### *2.4. Performance Evaluation Index and SHAP Explanation Model*

#### 2.4.1. Performance Evaluation Index

The classified river entities were labeled, with the main stream river reach set to 1 and the tributary river reach set to 0. The true positive (TP), false positive (FP), true negative (TN), and false negative (FN), as well as the confusion matrix, accuracy, precision, recall, F1_Score, and ROC-AUC curves, were used as model performance evaluation metrics.

Accuracy measures the proportion of correct classifications, precision refers to the proportion of correctly predicted positive samples in the detection framework, recall is the probability of actual positive samples being predicted as positive for the original samples, and the F1-score considers both accuracy and recall, finding a trade-off between the two. The ROC-AUC curve is formed by the True Positive Rate (TPR) and the False Positive Rate (FPR), while the AUC indicates the size of the area under the ROC curve. The higher the AUC, the more likely it is that the current classification algorithm will produce more positive samples than negative ones, resulting in better classification results. The ROC-AUC curve can effectively eliminate the influence of sample category imbalances on the index results. The above metrics are expressed as follows:

$$Acc = \frac{TP + TN}{TP + TN + FP + FN} \tag{3}$$

$$Pre = \frac{TP}{TP + FP} \tag{4}$$

$$Rec = \frac{TP}{TP + FN} \tag{5}$$

$$F1\_Score = \frac{2Pre \cdot Rec}{Pre + Rec} \tag{6}$$

where $P$ (positive) denotes predicted positive samples, $N$ (negative) denotes predicted negative samples, $T$ (true) denotes a correct prediction, and $F$ (false) denotes an incorrect prediction. Thus, $TP$ and $TN$ denote the number of correctly classified main stream and tributary samples, respectively, and $FP$ and $FN$ denote the number of incorrectly classified river main stream and tributary samples, respectively.

#### 2.4.2. SHAP Explanation Model

For structured data and classification tasks, ensemble learning methods generally learn well but have been unable to solve the interpretability problem; that is, for a specific sample, it is impossible to understand the impact of the sample's feature values on the result. Explainable artificial intelligence (EAI) aims to help people understand how models make decisions in the prediction process and provide guidance for feature selection and model optimization by improving model transparency. In this study, the Shapley additive explanation method (SHAP) from game theory was selected to interpret the Cat-Boost classification model and determine the importance of an individual by calculating the contribution of that individual to cooperation. SHAP is based on the Shapley value

explanation, which is an additive feature attribution method, and the predicted values of the model are interpreted as a linear function of the binary variables.

$$g(z') = \varphi_0 + \sum_{i=1}^{M} \varphi_i z_i'$$

(7)

where $M$ is the number of features in the feature vector $z'$; $z_i'$ is the mapping of the $i$-th feature $z'^{z_i' \in \{0,1\}}$; $\varphi_0$ denotes the model baseline value; and $\varphi_i \in R$ is the SHAP value of the feature $z_i'$ which represents the average marginal contribution to the predicted value of the model.

The larger the absolute value of SHAP, the greater the influence of the feature on the predicted value of the model, and its positivity or negativity value represents the direction of influence. The importance of each feature was measured according to the average absolute value of the SHAP value, calculated as follows:

$$I_j = \frac{\sum\limits_{n=1}^{N} \left| \varphi_j^{(n)} \right|}{N}$$

(8)

where $I_j$ denotes the importance of the $j$-th feature, $N$ denotes the number of samples, and $\left| \varphi_j^{(n)} \right|$ denotes the absolute value of the SHAP value corresponding to the jth feature in the nth sample.

The $i$-th sample is $x_i$, the $j$-th feature of the $i$-th sample is $x_{ij}$, the model's predicted value for that sample is $y_i$, and the baseline for the entire model (typically the mean of the target variable of all samples) is $y_{base}$. The SHAP value obeys the following:

$$y_i = y_{base} + f(x_{i1}) + f(x_{i2}) + \cdots f(x_{in})$$

(9)

where $f(x_{ij})$ is the SHAP value of $x_{ij}$ and $f(x_{in})$ is the contribution of the $n$-th feature in the $i$-th sample toward the final predicted value of $y_i$. The SHAP value for each feature indicates the expected change in model prediction when conditioned on that feature. When $f(x_{in}) > 0$, the feature boosts the prediction value; otherwise, the feature reduces the feature contribution.

## 3. Design and Acquisition of Main Stream and Tributary Cases

### 3.1. Main Stream and Tributary Case Design

In this study, a ternary representation was used to describe the mainstreams and tributaries of the cases; that is, three elements of each case reach are operated. Each river reach consists of a case object, case feature, and case classification label, expressed as follows:

$$CASE : \{O, F, L\}$$

where the case object ($O$) indicates the unique ID number of the river reach, the case feature ($F$) contains several quantitative indicators to describe the attributes, geometry, and relationships of the river reach, and the selection and expression of the features are the focus and difficulty of the case design. The case classification labels ($L$) consist of the main stream (1) and tributaries (0).

### 3.2. Main stream and Tributary Case Acquisition

The steps to obtain the main stream and tributaries of the case-base are as follows:

Step 1: Preprocessing. 1. Select case results. Select the graded river network from the expert case library and perform a topological check to remove surrounding small and scrambled rivers and those without sources to ensure the integrity and connectivity of the rivers. 2. Determine the flow direction of the river. Check river flow directions and manually correct rivers with incorrect flow directions according to the location of river

sources and outlets. 3. Divide river reach. Segmentation at intersections, and use the river reaches as the basic unit to obtain the feature metrics.

Step 2: Build the tree structure. Based on the data and graphical visualization, determine whether the river network structure satisfies the tree structure, digitize all boundary points, and construct a tree index by determining the downstream river reach ID of the river reach.

Step 3: Case feature calculation and identification. 1. Calculate feature values. Traverse all river reaches in the tree structure and calculate the feature index values in Table 1 for each river reach. 2. Mark the main stream and tributaries. Based on existing classification results, determine whether upstream reaches from the mouth of the river are of the same grade as the current river reach and mark those that are the same as the main stream (1) and those that are different as tributaries (0).

Most of the existing studies focus on the tree-shaped river network, while the trellis river network also has obvious characteristics of main stream and tributaries. Therefore, considering the diversity and adaptability of river networks with different shapes, trellis river networks were selected from the Open Street Map (OSM) [52] as the experimental data set. China's Min River network is a typical trellis river network in Fujian Province, China. Its geographic location is shown in Figure 1, and the area of its catchment is more than 60,000 square kilometers. The data classified by experts for this basin were selected as data sources for main stream and tributary cases. Some of the graded cases and main stream and tributary classification data are represented by different colors and widths in (Figure 2). Using the above case acquisition method, a total of 393 main stream and tributary cases were obtained, consisting of 197 main stream and 196 tributary cases, some of which are presented in Table 2.

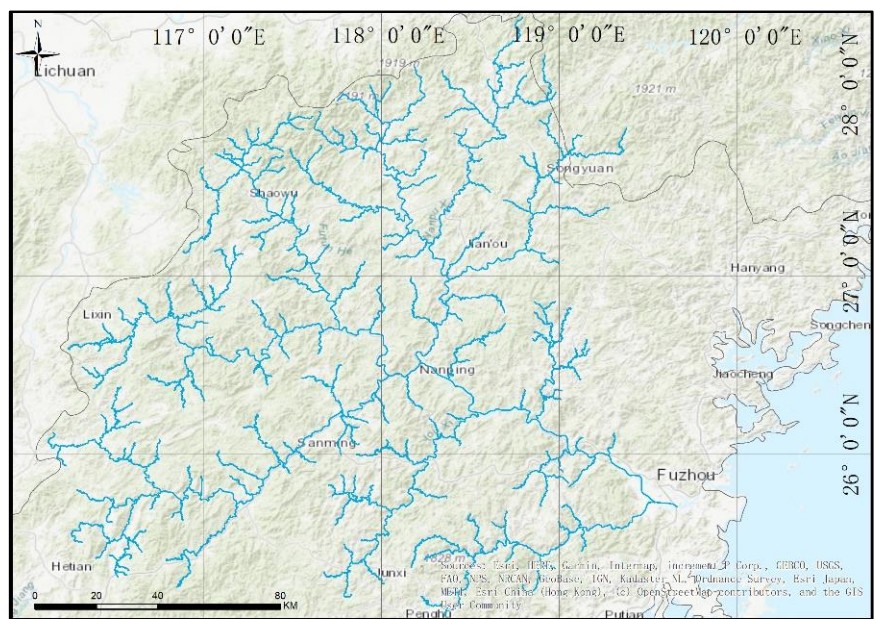

**Figure 1.** The map of Min River.

**Table 2.** Selected cases of main stream and tributary (part).

| Object (O) | Case Features (F) | | | | | | | | Label (L) |
|---|---|---|---|---|---|---|---|---|---|
| | Consistency of Semantic | Curvature of Reaches | Length of Reaches/m | Angle of Intersection/° | Number of Upstream Reaches | Depth of Reaches | Number of Upstream River Sources | Maximum Upstream Length/m | |
| ID_1 | 1 | 1.08 | 6248 | 158.16 | 2 | 9 | 12 | 155,702 | 1 |
| ID_2 | 1 | 1.31 | 12,111 | 155.49 | 2 | 13 | 27 | 184,395 | 1 |
| ID_3 | 0 | 1.34 | 4317 | 92.16 | 2 | 9 | 12 | 98,898 | 0 |

**Table 2.** *Cont.*

| Object (O) | Case Features (F) | | | | | | | | Label (L) |
|---|---|---|---|---|---|---|---|---|---|
| | Consistency of Semantic | Curvature of Reaches | Length of Reaches/m | Angle of Intersection/° | Number of Upstream Reaches | Depth of Reaches | Number of Upstream River Sources | Maximum Upstream Length/m | |
| ID_4 ⋮ | 1 | 1.36 | 20,890 | 119.19 | 0 | 1 | 1 | 0 | 1 |
| ID_157 | 1 | 1.59 | 25,972 | 123.75 | 0 | 1 | 1 | 0 | 0 |
| ID_158 ⋮ | 0 | 1.30 | 13,577 | 140.94 | 2 | 23 | 41 | 302,935 | 1 |
| ID_264 ⋮ | 1 | 1.60 | 46,350 | 146.06 | 0 | 1 | 1 | 0 | 0 |

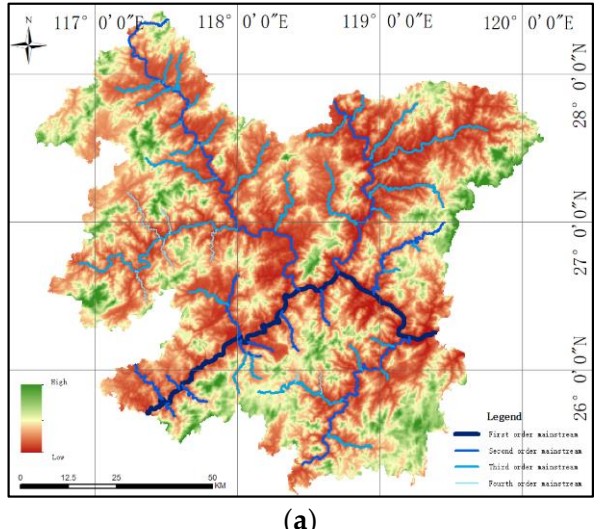

(**a**)

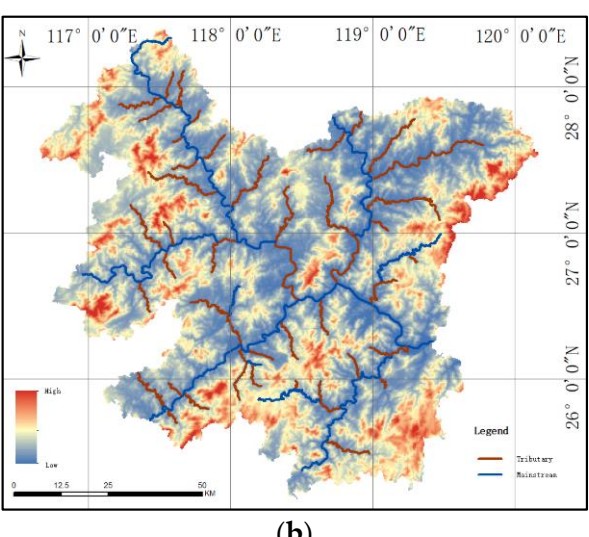

(**b**)

**Figure 2.** River network classification and case study of main stream and tributary. (**a**) Hierarchical display and (**b**) Main stream and tributary display.

## 4. Model Training and Interpretation

### 4.1. Model Training

The main stream and the tributary cases obtained in Section 3.2 were used as training samples, and the training samples were divided into two parts: 70% were used as the training set with a total of 275, while the remaining 30% were used as the test set with a total of 118, and the CatBoost classification model was constructed by training it. The Cat-Boost model has many hyperparameters, which are difficult to optimize simultaneously in practice. Therefore, five hyperparameters that significantly impacted the classification model were selected for grid search optimization. Among them, Learning_Rate controls the amplitude of the weight update in each iteration. A smaller learning rate can make the model more stable, while a larger learning rate may cause the model to fail to converge. To balance the accuracy and training time of the model, 0.03 and 0.1 were selected for search, respectively. Depth limits the depth of the decision tree. Increasing the depth of the tree can improve the complexity and fitting ability of the model, but it may lead to overfitting. According to the complexity of the data set and the number of samples, the depths of 4, 6, and 8 are selected for the search. L2_Leaf_Reg controls the complexity of leaf nodes by applying L2 regularization. Larger regularization parameters will promote the model to generate simpler trees, and we selected 3 and 5 for search according to the complexity of the data set. The iterations number determines the number of decision trees when training the model. More iterations can improve the accuracy of the model but also increase the training time. 100 and 1000 iterations were chosen according to the number of data sets. Scale_Pos_Weight mainly solves the problem of class imbalance. By adjusting the weight

scaling parameters of positive cases, the importance of positive and negative cases can be balanced, and the classification ability of the model for the target class can be improved. 0.05 and 1.0 were selected. The selected parameters are summarized in Table 3.

**Table 3.** Search space of parameters to be optimized and optimization results.

| Number | Name | Search Space | Optimization Results |
|---|---|---|---|
| Parameter 1 | Learning_Rate | (0.03, 0.1) | 0.03 |
| Parameter 2 | Depth | (4, 6, 8) | 4 |
| Parameter 3 | L2_Leaf_Reg | (3, 5) | 3 |
| Parameter 4 | Iterations | (100, 1000) | 100 |
| Parameter 5 | Scale_Pos_Weight | (0.05, 1.0) | 1.0 |

The trained model was tested using the test set, where a higher TPR and lower FPR in the ROC-AUC curve (Figure 3) indicate that the classification model has a higher sensitivity and represents excellent classification performance. The test set results and the statistics of the classification effectiveness metrics of the trained CatBoost model are listed in Table 4, which shows that the model achieved good results with a good fit, good learning, and excellent generalization.

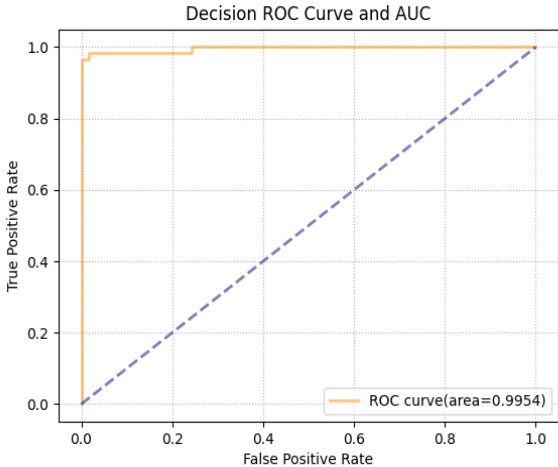

**Figure 3.** ROC-AUC curve of classification model.

**Table 4.** CatBoost classification effect index statistics.

| Test Set Result | TN 62 | FP 0 | FN 2 | TP 54 |
|---|---|---|---|---|
| | Accuracy (%) | Precision (%) | Recall (%) | F1-Score (%) |
| Training set | 98.91 | 100.00 | 97.87 | 98.93 |
| Test set | 98.31 | 100.00 | 96.43 | 98.18 |

### 4.2. SHAP Explanation

As a tree integration model, CatBoost uses Tree Explainer, a tree model explanation in the SHAP explanation model, to explain the CatBoost classification model. The Tree Explainer calculates the contribution of each feature to the prediction on each sample and weights the average of these contribution values to obtain the SHAP value of each feature. These SHAP values represent the degree of influence each feature has on the model prediction and are used to interpret the prediction results of the model. It includes global and local explanations. Global explanations, that is, the effect of features on the overall model, can be used as feature importance to help filter variables. In this paper, both single and

interaction feature maps are visualized, which not only consider the effect of individual variables but also the synergistic effect between variables. The local interpretation, that is, the interpretation of the prediction results of a single sample, allows one to visualize the primary influential features of the prediction results in a single sample and the degree of influence of the corresponding features.

### 4.2.1. Global Explanation

(1)　Single-feature explanation

The SHAP summary plot shows the distribution of each feature and the degree to which each feature affects the model output. Figure 4 shows a scatter diagram of the SHAP values of each sample feature, where the $x$-axis refers to the feature influence weight and the $y$-axis is arranged in descending order of the feature importance. The wider the distribution area, the greater the influence of the feature on the classification result. The region distribution of consistency of semantic (C_S) in the figure is the widest; that is, it has the most significant influence on main stream and tributary classification. Each point represents a sample feature, and different colors indicate the effect of the size of the feature value on the results (red for high, blue for low, and purple for close to the mean). For example, from the color distribution of the points of the semantic consistency feature, it was found that the smaller the feature value, the smaller the SHAP value, and the larger the feature value, the larger the SHAP value. Therefore, the value of the feature is positively correlated with the SHAP value.

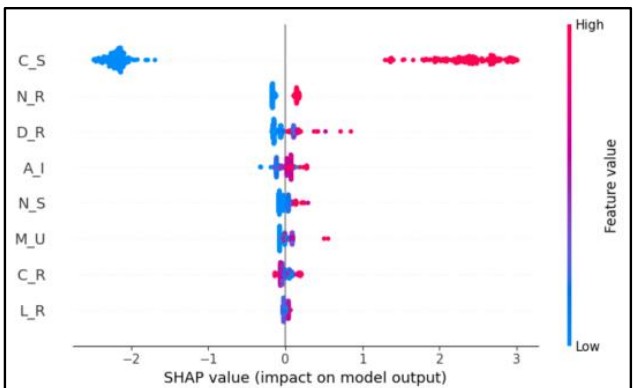

**Figure 4.** SHAP summary. (Note: For the convenience of display in the figure, all features in Figures 4–9 are abbreviated).

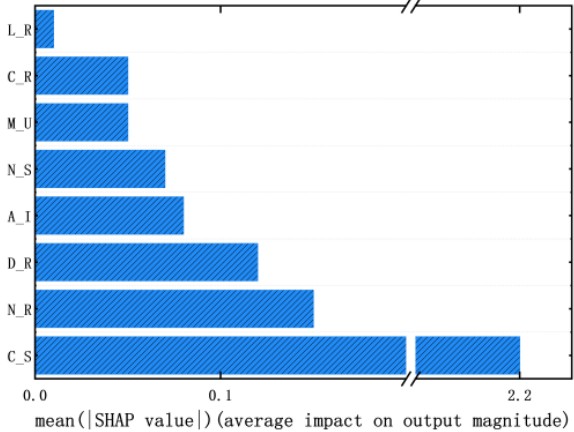

**Figure 5.** SHAP feature importance.

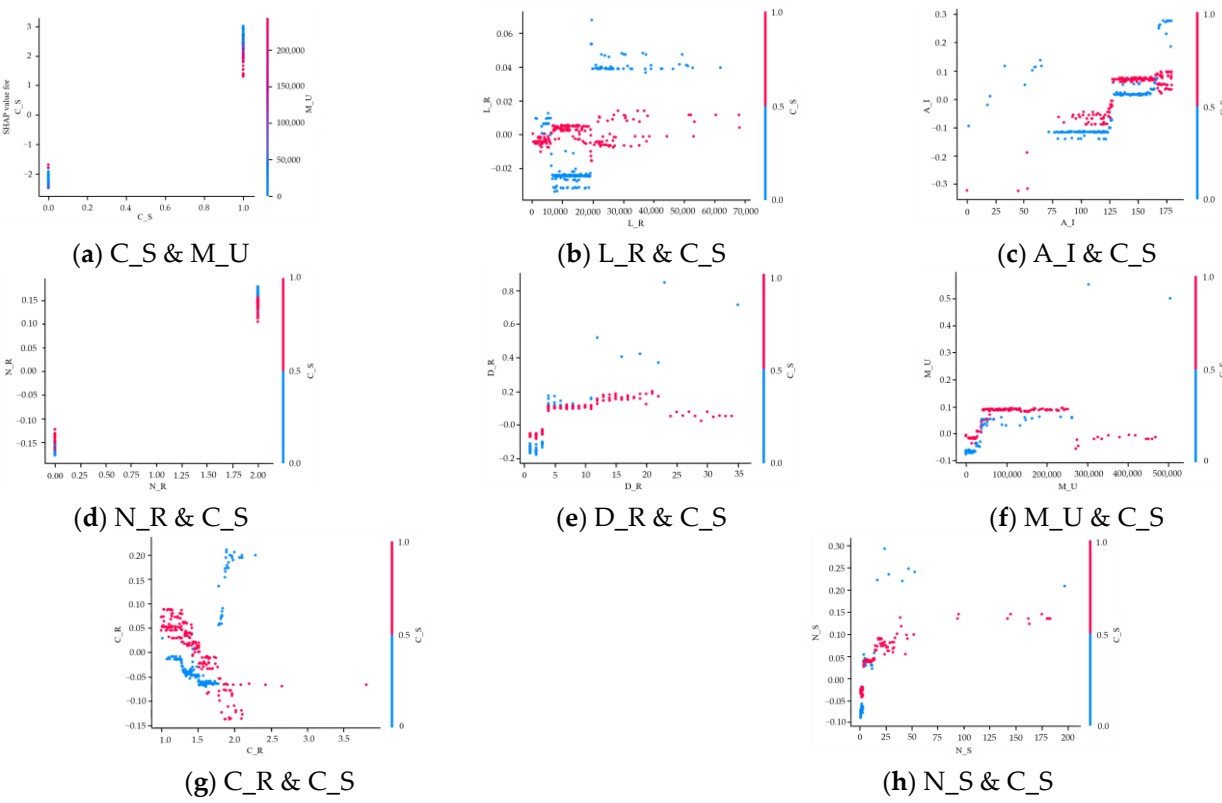

**Figure 6.** SHAP feature interaction dependence.

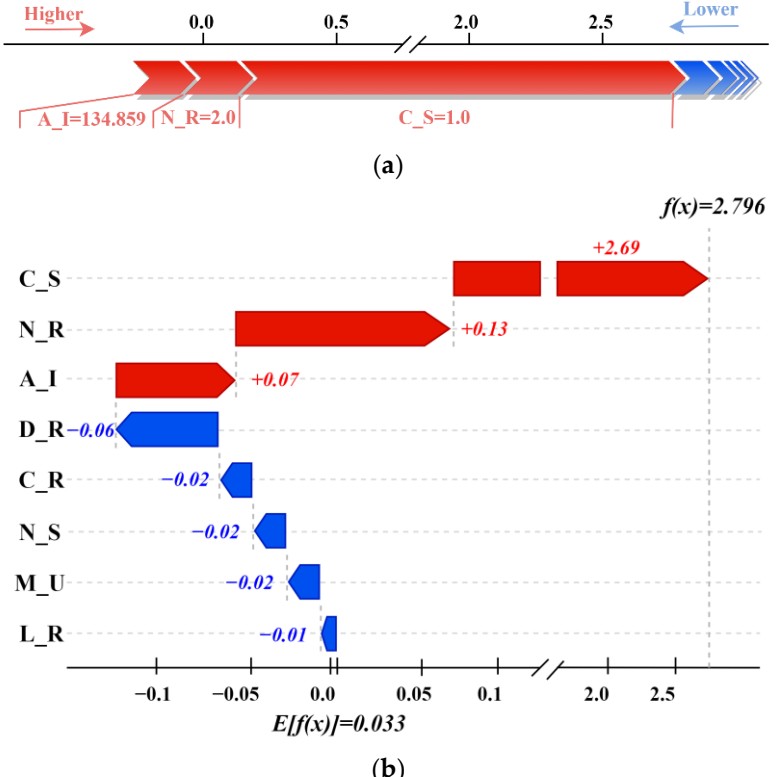

**Figure 7.** Partial interpretation for main stream sample. (**a**) Force Plot interpretation for main stream sample and (**b**) Water Plot interpretation for main stream sample.

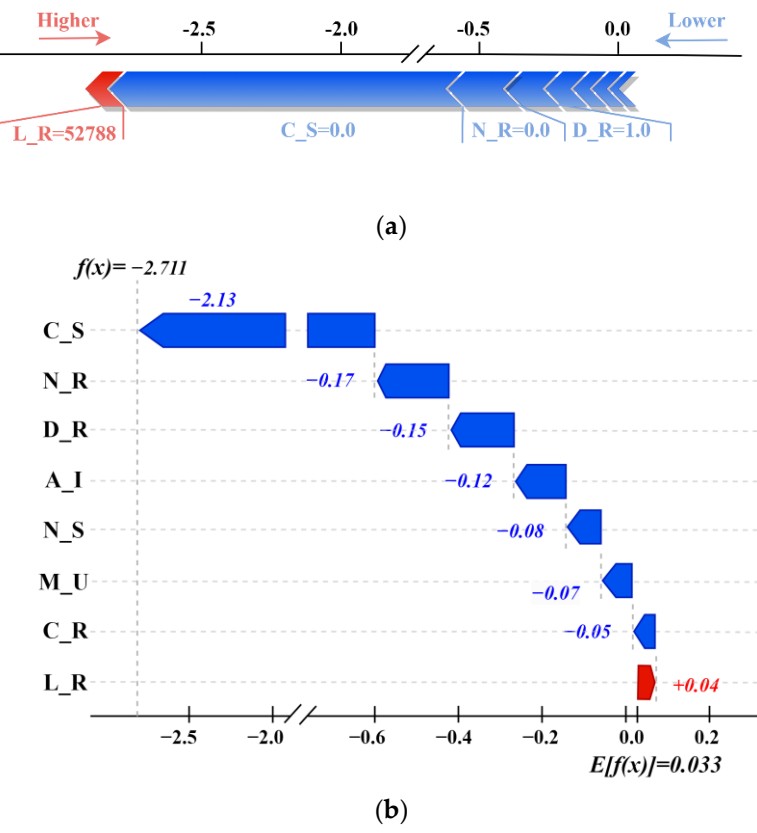

(**a**)

(**b**)

**Figure 8.** Partial interpretation for tributary sample. (**a**) Force Plot interpretation for tributary sample and (**b**) Water Plot interpretations for tributary sample.

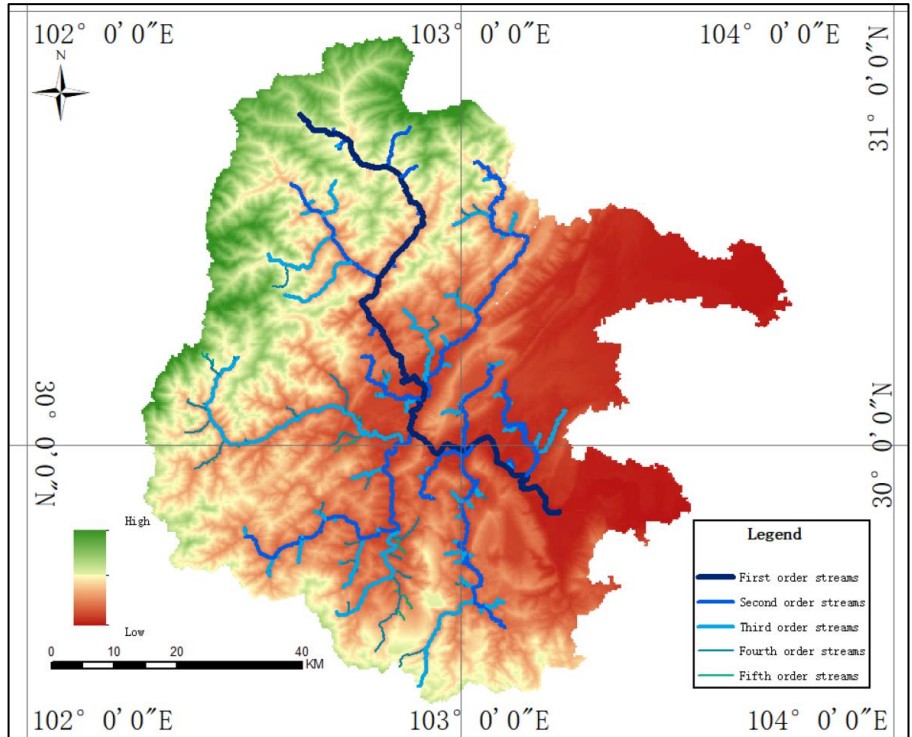

**Figure 9.** Grading results of river network.

The SHAP feature importance plot is used to display the relative importance of each feature to the model output. Figure 5 shows a bar chart based on the average absolute value of the SHAP value for each feature, showing the global importance of the features. The ranking of the quantified feature importance in Figure 5 is consistent with that shown in Figure 4. The top-ranked feature is the consistency of semantics; with a SHAP value greater than 2, its importance area is 10–200 times that of the other features, which shows that its importance is much greater than that of the rest of the features. Furthermore, the topological features with relatively high importance were the number of upstream reaches and depth of reaches, with SHAP values greater than 0.1. The importance of the remaining features was lower, and the geometric feature length of the reaches had the least influence. As shown in Figures 4 and 5, consistency of semantic has the highest impact on the classification results, and the accuracy of the classification results is reduced if the feature is extracted separately and used for classification. Therefore, although this feature has a high impact on classification, it can achieve a better classification effect when applied alongside other features.

(2)  Partial dependence

The feature interaction dependence plot can intuitively display the interaction relationship between different features in the model, thereby better understanding how the model combines features to make predictions (Figure 6). Here, the *x*-axis represents the actual value of the feature value, and the *y*-axis represents the SHAP value of the feature, which indicates the degree to which the feature value changes the predicted model output. To show the driving interaction effect between features, the second feature is colored by the second feature in this feature-dependent scatter plot (the default second feature is the automatically selected column of features with the strongest interaction with this feature), indicating the distribution of the second feature during the change in this feature. In Figure 6, the feature with the strongest interaction with the consistency of the semantic feature was the maximum upstream length, indicating that when combined with the maximum upstream length, the consistency of the semantic has a more significant impact on the model output than other feature combinations.

The driving effects of each feature in Figure 6 can be analyzed as follows:

(1)  In Figure 6a, the consistency of the semantics of value one contributes positively to the model predictions. This has a greater impact on the classification of the absolute value of the features when the maximum upstream length feature value is low.

(2)  In Figure 6b, the value of the length of reaches is in the range of 5000–20,000, which contributes negatively to the model predictions. It has less impact on classification when the value of the consistency of the semantic feature is one.

(3)  In Figure 6c, the model predictions tend to be stable when the angle of intersection is >75. There was little difference in the impact on classification, regardless of whether the consistency of the semantic feature value was 1 or 0. When the feature value is <75, it has less impact on the classification when the consistency of the semantic value is 1.

(4)  In Figure 6d, the number of upstream reaches of value 2 positively contributes to the model prediction value. Like in (a), when the consistency of the semantic feature value is zero, the impact on the feature absolute value classification is greater.

(5)  In Figure 6e, the larger the depth of reaches, the larger the feature value. When the depth exceeds five, it contributes positively to the model's predicted value. When the consistency of the semantic feature value is 1, it has a greater impact on classification.

(6)  In Figure 6f, if the maximum upstream length is >50,000, it contributes positively to model prediction, with a peak at approximately 250,000. Most samples significantly impact the classification when the consistency of the semantic feature value is 1.

(7)  In Figure 6g, the larger the value of the curvature of the reaches, the smaller the corresponding feature value. When the consistency of the semantic feature value is 1 or 0, it has less impact on classification.

(8) In Figure 6h, when the number of upstream river sources is in the range of 0–50, the higher the number of river sources, the higher the feature value, with an increasing trend, which contributes positively to the model prediction value when it is greater than 50. When the consistency of the semantic feature value is zero, it has a greater influence on the classification. When the consistency of the semantic feature value is 1, it has a smaller but more stable influence on the classification results.

4.2.2. Partial Explanation

The main stream with ID = 346 and tributary with ID = 275 were selected separately to explain the CatBoost classification model from the perspective of a single sample. Force and water plots were calculated and visualized for the main stream and tributary samples. In the force plot, each feature value is a force that increases or decreases the prediction, and each attribute value is an arrow that indicates an increase or decrease in the prediction, with red and blue arrows representing a positive and negative contribution, respectively; the higher the length, the more significant the contribution. In the water plot, the horizontal axis indicates the SHAP value, and the vertical axis indicates the value of each sample feature. Blue indicates that the feature harms the prediction, and an arrow pointing to the left indicates a decrease in the SHAP value. Red indicates that the feature positively affects the prediction, and an arrow pointing to the right indicates an increase in the SHAP value, where E[f(x)] denotes the SHAP base value, which is the mean value predicted by the model.

In the main stream sample explanation plot (Figure 7), the consistency of semantics, angle of intersection, and number of upstream reaches play a positive driving role, and the consistency of the semantic value of 1 has the most significant positive predictive effect on the model. The final output SHAP value was 2.796, which is much larger than the base value of 0.033. The sample was predicted to be positive, i.e., mainstream.

In the tributary sample explanation plot (Figure 8), the length of the reaches had a positive promotional effect on model prediction. However, it was much smaller than the sum of the negative inhibitory effects of the other features of the model. The consistency of the semantic value of zero had the most significant negative inhibitory effect on the model. The final output SHAP value was −2.711, which is smaller than the baseline value of 0.033. Thus, the sample was predicted to be negative, i.e., a tributary.

## 5. Testing and Analysis of the Results

To verify the effectiveness and practicality of the classification method, the results of this study using the CatBoost algorithm were compared with the classification results of three standard machine learning classification methods, namely, CART decision tree [53], random forest [54], and logistic regression [55], as well as three boost classification algorithms, namely, Adaptive Boosting (AdaBoost) [56], GBDT [57], and eXtreme Gradient Boosting(XGBoost) [58,59] (Table 5). Among them, CART decision tree is an algorithm based on tree structure for decision-making; random forest is an algorithm based on ensemble learning of multiple decision trees; logistic regression is an algorithm based on probabilistic models for classification; AdaBoost is an algorithm based on weighted weak classifiers for iteration; GBDT is an iterative decision tree algorithm based on gradient descent; and XGBoost is an improved algorithm based on GBDT. The boost algorithms outperformed conventional classification algorithms based on an overall comparison of the two types of models. Of the conventional machine learning classification algorithms, random forest performed the best, and logistic regression performed the worst. Among the boost classification algorithms, CatBoost had the highest accuracy (Acc), precision (Pre), and F1 scores. Compared with the other classification algorithms, the Acc, Pre, and F1-score results were 0.85–5.94%, 1.82–9.84%, and 0.8–5.74% higher, respectively. On the one hand, the method proposed in this study can make better use of the data characteristics of the data set and can effectively process the classification features. On the other hand, through a reasonable selection of hyperparameters, the model achieves better performance.

Therefore, the results show that the method proposed in this study can successfully classify mainstreams and tributaries.

**Table 5.** Comparison of evaluation indexes of model classification results.

| Classification Method | Correct Classification Quantity | | Wrong Classification Quantity | | Score (%) | | | |
|---|---|---|---|---|---|---|---|---|
| | TN | TP | FN | FP | Acc | Pre | Rec | F1-s |
| CatBoost | 62 | 54 | 2 | 0 | 98.31 | 100.00 | 96.43 | 98.18 |
| CART decision tree | 59 | 54 | 2 | 3 | 95.76 | 94.74 | 96.43 | 95.58 |
| Random forest | 63 | 52 | 2 | 1 | 97.46 | 98.11 | 96.30 | 97.20 |
| Logistic regression | 54 | 55 | 3 | 6 | 92.37 | 90.16 | 94.83 | 92.44 |
| Adaboost | 59 | 55 | 0 | 4 | 96.61 | 93.22 | 100 | 97.30 |
| GBDT | 55 | 60 | 1 | 2 | 97.46 | 96.30 | 98.11 | 97.20 |
| XGBoost | 59 | 56 | 0 | 3 | 97.46 | 98.18 | 96.43 | 97.30 |

The proposed method in this study was used to identify the main stream and tributaries of an ungraded river network. The dataset was derived from OSM, which was also a trellis river network. After pretreatment, topological check, and feature extraction, the river network has 303 river reaches. Some of the calculation processes and classification results are listed in Table 6. Given that the identification of mainstreams and tributaries is relative, only one of two or more upstream reaches of the same river that best matches the characteristics of the main stream was identified as the mainstream, and the remainder were classified as tributary reaches. Calculate the probability of the river reaches being divided into mainstreams and tributaries, and determine the river reach with a higher probability of main stream as the mainstream. However, when the river reach at the intersection is of the same type, the main stream probability of the two is compared, and the one with a higher main stream probability is divided into the main stream and the other as a tributary. This classification method can be used to determine the classification results for river reaches with unclear main stream and tributary classifications by comparing classification probabilities, with the river reach with the highest probability of being upstream main stream at each intersection point determined to be the main stream river reach. For example, the river reaches with ID = 45 and ID = 272 have common downstream reaches, and their main stream probabilities are 0.94706 and 0.94217, respectively. Thus, ID of 45 was determined to be the main stream reach.

The river network was graded according to the above classification results (Figure 9). The classification in this study considered the semantic, geometric, and topological features of river networks, making the identification of the mainstreams and tributaries of river networks more effective, and the main stream features are noticeable after classification. It can be observed from the graded rivers that the river network has a rational hierarchical structure. The main stream and tributary relationships are apparent between all levels of rivers, and the parent–child and left-right branch relationships are clear, which is more consistent with manual identification results and satisfies the mapping requirements.

**Table 6.** Experimental results (partial).

| Object ID | Lower Reach ID | Consistency of Semantic | Curvature of Reaches | Length of Reaches | Angle of Intersection | Number of Upstream Reaches | Depth of Reaches | Number of Upstream River Sources | Maximum Upstream Length | Classification Probability of Mainstream | Classification Probability of Tributary | Final Classification Result |
|---|---|---|---|---|---|---|---|---|---|---|---|---|
| ID = 45 | ID = 48 | 1 | 1.12 | 1273.02 | 165.74 | 2 | 6 | 8 | 55,095.77 | 0.94706 | 0.05294 | Mainstream |
| ID = 272 | ID = 48 | 1 | 1.61 | 5689.93 | 109.95 | 2 | 5 | 7 | 42,137.27 | 0.94217 | 0.05783 | Tributary |
| ID = 268 | ID = 125 | 0 | 1.36 | 14,805.74 | 169.30 | 2 | 15 | 20 | 78,901.16 | 0.21010 | 0.78990 | Mainstream |
| ID = 269 | ID = 125 | 1 | 1.55 | 15,427.66 | 50.45 | 2 | 9 | 12 | 60,214.88 | 0.91301 | 0.08699 | Tributary |

## 6. Discussion and Conclusions

### 6.1. Discussion

Although the proposed method has shown good performance in improving the accuracy of river network classification, there are still some limitations that need to be addressed in future research. For example, the method is highly dependent on the quality and completeness of input data, which may not perform well in areas with poor data availability. In addition, the method only considers the static features of river networks and does not take into account dynamic changes caused by natural or human factors. Finally, the scalability and adaptability of the method also need to be further explored to meet the classification requirements of river networks of different scales and in different regions. Therefore, in future research, efforts will be made to explore how to integrate the dynamic features of river networks and improve the robustness of the method under different data scenarios. By combining more high-quality classification cases and multiple indicators and evaluation systems, the algorithm and model structure will be further improved to enhance the accuracy and efficiency of the model. Although there are some limitations and shortcomings in the current method, it provides new ideas and methods for research and practice in the field of river network classification and has important research value and practical significance.

### 6.2. Conclusions

To further improve the accuracy of river network classification and evaluate the feature weight, this paper proposes an automatic grading method for river networks based on ensemble learning in CatBoost. Main stream and tributary cases and semantic, geometric, and topological feature terms were obtained from existing river network classification results. A classification model was obtained by training the CatBoost algorithm. The SHAP framework was introduced to interpret the classification model and determine the degree of influence of different feature values on the classification results. Finally, a hierarchical classification of the river network was conducted. The main conclusions are as follows:

(1) In this study, semantic, geometric, and topological features were used to describe river network characteristics, and features such as intersection angle and curvature of river reaches were introduced to describe the main stream and tributary characteristics according to their unique characteristics. This ensures that the classification model can efficiently identify the main stream and tributary river reaches while maintaining the characteristics of the river reaches themselves and ensuring the structural relationship between river networks. Thus, automatic classification was realized.

(2) By enhancing the feature engineering and optimizing the hyperparameters, the CatBoost model performs well in the case-base river network classification task. On the one hand, by exploring and mining the dataset, the river network is constructed into a binary tree, the internal topological relationship of the river reach is extracted, and the semantic and geometric features are combined to make the model better use of the information in the data, thus improving the classification performance of the model. On the other hand, by optimizing the hyperparameters, the optimal parameter combination can be found, which further improves the accuracy of the model.

(3) Case learning based on expert experience makes good use of existing grading experience and knowledge, and the CatBoost classification method solves the difficulty associated with determining weights and thresholds for multiple indicators. Compared with the other classification algorithms, the accuracy, precision, and F1-score were better by 0.85–5.94%, 1.82–9.84%, and 0.8–5.74%, respectively. Therefore, automatic river network classification intelligence was improved using this method.

(4) The descriptive analysis of the model using the SHAP explanatory framework showed that the consistency of semantic features had the most significant influence on the classification results, but the classification accuracy when combined with other features was higher than when used alone. The second most influential factor was the depth of the reaches, followed by number of upstream reaches and the maximum

upstream length. The geometric feature length of the reaches had the most negligible effect on the classification results. The value of semantic consistency positively drives the model classification results and has the most robust interaction with the other features.

**Author Contributions:** Di Wang composed the original idea of this study, implemented the code, contributed to writing, and finished the first draft. Haizhong Qian supervised the research and edited and reviewed the manuscript. All authors have read and agreed to the published version of the manuscript.

**Funding:** This research was funded by the Natural Science Foundation for Distinguished Young Scholars of Henan Province, grant number [212300410014]; and the National Natural Science Foundation of China, grant number [42271463] [42101453] [42371461].

**Data Availability Statement:** The data that support the findings of this study are available in Available online: https://github.com/96-WD/GiRiver_Classification (accessed on 1 September 2023).

**Acknowledgments:** The authors would also like to thank Peixiang Duan and Longfei Cui for their theoretical and technical assistance, which significantly improved the quality of this study.

**Conflicts of Interest:** The authors declare that they have no known competing financial interest or personal relationships that could have influenced the work reported in this study.

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
