# Peer review of "CatBoost-Based Automatic Classification Study of River Network"

_ijgi, doi:10.3390/ijgi12100416_

Round 1

Reviewer 1 Report

This article proposes a river network classification algorithm based on the CatBoost model and conducts experimental verification. A SHAP framework was also proposed to explain the experimental results. Overall, the work of this article has certain practical value. However, before the paper can be published, the author still needs further revisions and improvements.

The specific comments are as follows:

1) The innovation points of this article need to be further refined, especially regarding the CatBoost model. How did the author improve it?

2) Is the author considering including SHAP in the title, as this is also an important work of this article. At present, it cannot be reflected from the title.

3) The abstract only reflects the accuracy of the experiment, has there been any improvement in efficiency?

4) I think the experimental data should be a lack of rigor in this article. Generally, for algorithm validation, public datasets or detailed explanations of the data are used, but this article provides too little information. It is recommended to supplement.

5) The paper lacks a discussion section.

6) All thematic maps, such as figure 2, lack the basic elements of cartography, such as the north arrow, scale, etc.

7) Due to formatting issues, some table headers are misaligned, seriously affecting reading. As shown in Tables 2, 6, etc.

8) The font size of some images is too small, which affects reading. As shown in Figure 6.

9) On page 5, section 2.3.2, "Explainable artistic intelligence (XAI)", is it "EAI"?

Minor editing of English language required.

Reviewer 3 Report

The abstract doesn't provide sufficient context or background information about the importance of performance evaluation in river network classification. Some context about the significance of performance metrics in this specific domain would make the section more informative and relevant.

While the abstract mentions using accuracy, precision, recall, F1_Score, and ROC-AUC curves as performance metrics, it doesn't provide any specific results or comparisons of these metrics. Including some results or comparisons with other methods would help readers understand the effectiveness of the proposed approach.

In some places, the authors uses vague or unclear language. For example, "automatic river network classification intelligence was improved using this method" could be more specific about how the intelligence was improved and by what degree.

Reviewer 4 Report

I have major concerns, and the details are as follows:

Abstract

·       "existing case results". Provide more context on these cases. Were these previous studies, datasets, or expert knowledge? This would help readers understand the foundation for the chosen features.

·        0.85%-5.94%, 1.82%-9.84%, 0.8%-5.74% add respectively.

·       While the abstract hints at the method's superiority over other algorithms, it could explicitly state the key contribution of the paper. This could be a unique approach, a novel combination of features, or a novel way of using CatBoost for river network classification.

·       Briefly explain what the SHAP framework is and why it's used in this context. This would provide readers unfamiliar with SHAP with a better understanding of its role in the methodology.

·       Some sentences could be rephrased for greater clarity. For instance, "The experiment proves that the method in this study imitates experts better" could be rephrased as "The experiment demonstrates that the proposed method achieves expert-level imitation."

Introduction

·       While you introduce various concepts such as river network selection, simplification methods, and the CatBoost algorithm, it might help to explicitly connect these concepts back to the overarching goal of river network classification. This will help readers understand how each component contributes to the final goal.

·       It might be helpful to include brief summaries of the key findings or contributions of these studies to highlight how your work builds upon or differs from them.

Main principles of the algorithm

·        Some parts of the section are repeated, particularly the description of CatBoost's innovations, which are mentioned multiple times. Consider consolidating these explanations for clarity and to avoid redundancy.

Design and acquisition of mainstem and tributary cases

·       When introducing technical terms like "ternary representation," ensure that their meanings are clearly defined or explained for readers who may not be familiar with the terminology.

·       When describing the steps to obtain mainstem and tributaries, consider breaking down each step into concise bullet points for clarity and ease of understanding.

·       In the final part, where you mention the Min River network in Open Street Map, it might be helpful to provide a brief context or description of why this specific dataset was chosen and how it relates to the research.

Model training and interpretation

·       Provide the actual numbers of samples in training and testing sets.

·       Ensure that specialized terms are well-defined or explained, especially when introducing technical terms like "SHAP value" and "Tree Explainer." This helps readers who might not be familiar with the concepts.

·       Consider formatting the subsection titles consistently to enhance readability. Additionally, breaking down the information using bullet points or shorter paragraphs could make the content more digestible.

·       As the reader, I can understand that certain features are impactful based on the provided analysis. To enhance comprehension, you could explicitly highlight key insights, such as which features contribute most significantly to the model's performance or how they interact with each other.

Testing and analysis of the results

·       The comparison of the proposed CatBoost algorithm with other classification methods provides valuable insights into its performance. It would be helpful to briefly explain what each of these comparison algorithms is (e.g., CART decision tree, random forest, logistic regression, etc.) to aid readers who might not be familiar with these terms.

·       You mention that the boost classification algorithms outperformed conventional ones, and CatBoost performed the best in terms of accuracy, precision, and F1-score. It would be beneficial to provide a bit more context about why CatBoost might have performed better. Was it due to handling categorical features more effectively, optimizing hyperparameters, or another factor?

·       The explanation of how the classification method works when identifying mainstems and tributaries based on classification probabilities and the comparison of mainstem probabilities could be more detailed. This process is crucial and ensuring that readers clearly understand how the method works would improve comprehension.

·       Conclude this section by summarizing the key findings from the testing and analysis. Emphasize how the proposed CatBoost method's performance aligns with the study's objectives and why it is particularly effective for this application.

Conclusions

The conclusions effectively summarize the study's goals, methodology, findings, and the significance of the proposed approach in enhancing river network classification accuracy. They succinctly capture the essence of the study and its implications.

In general

·       Some figures are poor and must be improved.

·       Some parts of the text appear dense and could benefit from breaking down complex sentences into more digestible ones. This would improve overall readability and comprehension.

·       Consider adding more transition sentences between different aspects of your explanation. These sentences can help guide the reader smoothly from one concept to another and maintain a logical flow.

·       Some of the sentences could be made more concise while retaining the necessary information. This will help keep the reader engaged and avoid overly complex sentence structures.

·       Some sentences are quite complex and may benefit from being broken down into smaller, more straightforward sentences to improve readability.

Good

Round 2

Reviewer 4 Report

Figures are poor and must be improved.
